# Single CT Appointment for Double Lung and Colorectal Cancer Screening: Is the Time Ripe?

**DOI:** 10.3390/diagnostics12102326

**Published:** 2022-09-27

**Authors:** Mario Mascalchi, Giulia Picozzi, Donella Puliti, Giuseppe Gorini, Paola Mantellini, Lapo Sali

**Affiliations:** 1Department of Clinical and Experimental Biomedical Sciences “Mario Serio”, University of Florence, 50100 Florence, Italy; 2Division of Epidemiology and Clinical Governance, Institute for Study, Prevention and Network in Oncology (ISPRO), 50139 Florence, Italy; 3Division of Screening, Institute for Study, Prevention and Network in Oncology (ISPRO), 50139 Florence, Italy; 4IFCA Hospital, 50139 Florence, Italy

**Keywords:** chest CT, computed tomography, CT colonography, colorectal cancer, lung cancer, screening

## Abstract

Annual screening of lung cancer (LC) with chest low-dose computed tomography (CT) and screening of colorectal cancer (CRC) with CT colonography every 5 years are recommended by the United States Prevention Service Task Force. We review epidemiological and pathological data on LC and CRC, and the features of screening chest low-dose CT and CT colonography comprising execution, reading, radiation exposure and harm, and the cost effectiveness of the two CT screening interventions. The possibility of combining chest low-dose CT and CT colonography examinations for double LC and CRC screening in a single CT appointment is then addressed. We demonstrate how this approach appears feasible and is already reasonable as an opportunistic screening intervention in 50–75-year-old subjects with smoking history and average CRC risk. In addition to the crucial role Computer Assisted Diagnosis systems play in decreasing the test reading times and the need to educate radiologists in screening chest LDCT and CT colonography, in view of a single CT appointment for double screening, the following uncertainties need to be solved: (1) the schedule of the screening CT; (2) the effectiveness of iterative reconstruction and deep learning algorithms affording an ultra-low-dose CT acquisition technique and (3) management of incidental findings. Resolving these issues will imply new cost-effectiveness analyses for LC screening with chest low dose CT and for CRC screening with CT colonography and, especially, for the double LC and CRC screening with a single-appointment CT.

## 1. Introduction

The 2021 statements of the United States Prevention Service Task Force (USPSTF) have recommended both chest low-dose computed tomography (LDCT) for Lung Cancer (LC) screening and CT colonography for colorectal cancer (CRC) screening [1,2]. This justifies consideration of the opportunity for a single-appointment CT for double screening of LC and CRC. However, importantly, while chest LDCT is the only recommended screening tool for LC since 2013, other screening tools besides CT colonography have also been recommended for CRC since the late 1990s [1,2].

In the present article, we review the available evidence concerning the epidemiology and pathology of LC and CRC, and operational features, costs, harm and cost-effectiveness analyses of the chest LDCT and CT colonography, each considered as a stand-alone screening intervention. Then, we focus on the possibility of combining chest LDCT and CT colonography in a single appointment for double LC and CRC screening, showing how this approach appears feasible and is already reasonable as an opportunistic screening intervention.

Finally, we outline the persistent unresolved issues pertaining to chest LDCT and CT colonography that also apply to the single appointment CT approach for double LC and CRC screening.

## 2. Epidemiology

LC and CRC are among the most common and lethal neoplasms, accounting for about 2 and 1.8 million cases and 1.7 and 0.8 million deaths per year worldwide, respectively [3]. Accordingly, the overall 5-year survival rate of LC is 20.5% [4], and that of CRC is 73.7% [5]. Together, LC and CRC account for about 1/5 (21.8%) of all cancer cases and 1/4 (27.6%) of all cancer deaths [3].

Since early stage LC has a better prognosis and is more amenable to treatment than more advanced-stage LC [1], and removal of precancerous lesions halts the progression from polyp to CRC [2], screening of LC and CRC can save lives and is fully justified as a health preventive intervention. 

LC has sporadic distribution and cigarette smoking, a modifiable behavior, is its main key risk factor, along with age [1]. The proportion of deaths from LC attributable to smoking is 76% among males and 39% among females [6]. Additional risk factors for LC include second-hand smoke, environmental (radon, domestic fuel smoke, outdoor air pollution) and professional (asbestos, ionizing radiations, chromium, arsenic, etc.) exposures, infections and chronic inflammations, including Chronic Obstructive Pulmonary Disease (COPD) and Interstitial Lung Disease (ILD) [1,7,8,9,10]. Worldwide, 15–20% of men and around 50% of women with LC, particularly in Asia, are never smokers, whereas in the US, 9% of men and 19% of women with LC are never smokers [11,12].

Notably, several studies have demonstrated that selection of subjects to be invited to LC screening based on age and pack years only, as indicated by the USPSTF, performs worse in terms of predicting LC risk, and hence are suboptimal for subjects selection compared to personal risk prediction calculation models, which consider additional risk factors such as history of respiratory diseases, previous malignancy, family history of lung cancer (first-degree relative diagnosed at age 60 years or younger), and exposure to asbestos [13,14]. So far, the modified Liverpool Lung Project [LLPv2] and Prostate, Lung, Colorectal and Ovarian Cancer Screening Trial [PLCO] models [15,16] have been those more frequently utilized to select patients for LC screening in a clinical trial.

CRC too has a predominantly sporadic distribution, but familial history of the disease is present in 20–30% of cases, and rare inherited diseases such as Lynch Syndrome and Familial Adenomatous Polyposis and others are observed in 6–10% of cases, [17]. Age, race and history of inflammatory bowel disease (ulcerative colitis and Crohn’s colitis) are additional non-modifiable risk factors for CRC. However, several modifiable lifestyles and environmental risk factors are now also known for colon polyps and CRC [17], including diet—in particular, high intake of red and processed meat [18]—smoking, alcohol intake, little physical activity and elevated body mass index. Subjects with family history, the above inherited diseases or with inflammatory bowel disease are considered at “high-risk” of CRC and must be distinguished from the remaining subjects who are considered at “average-risk”.

Increased smoking habit is responsible for a growing trend of LC incidence and mortality in women, whereas these are decreasing in men [19]. An unexplained increase worldwide in rates of CRC in individuals under 50 has been recently described [17].

## 3. Pathology

In general, early stage tumors and precancerous lesions are the target of screening procedures. 

In the lung, only Non-Small Cell Lung Cancer (NSCLC) shows the relatively low growth compatible with screening, which affords its detection in early stages (I and II) when it can be surgically removed and definitely cured. Unfortunately, this is not the case for the fast-growing Small Cell Lung Cancer (SCLC), which accounts for 15–20% of primary lung tumors. SCLC escapes screening, and no benefit of its detection in the screening setting was reported in terms of decreased mortality [20]. Among the screen-detected NSCLC, adenocarcinoma and squamous carcinoma are the most frequent [21,22,23,24]. Adenocarcinoma is particularly frequent in women, and its long sojourn time [25] is associated with a more marked decrease in LC mortality following LDCT screening in women than in men [21,23,24,26]. 

Precancerous lesions in the lung can be distinguished into those related to adenocarcinoma and those related to squamous cell carcinoma. The former typically appear as a peripheral lung nodule, are easily demonstrated by chest LDCT and include (1) atypical adenomatous hyperplasia and (2) several distinct conditions that have replaced the old comprehensive term bronchoalveolar carcinoma (BAC) as adenocarcinoma in situ, minimally invasive adenocarcinoma, lepidic predominant adenocarcinoma and invasive mucinous adenocarcinoma [27]. The CT correlates of these histologic entities vary from lung nodules with pure ground glass (non-solid) density to mixed (part solid) and solid density (see below). Dysplastic lesions of the central airways which are precursors of squamous cell carcinoma are well detected by fluorescence bronchoscopy [28] and can escape chest LDCT. 

From a histological point of view, CRC is usually an adenocarcinoma (86% of all colon cancers- others include adenosquamous carcinoma, squamous cell carcinoma, spindle cell carcinoma, undifferentiated carcinoma and other special histopathological types) and arises from two precancerous lesions, namely adenomatous polyps, underlying 60–70% of CRC, and sessile serrated lesions, accounting for about 15–30% of CRC [17], which can coexist in the same subject, but have a distinct aspect at optical colonoscopy. In fact, at optical colonoscopy, adenomatous polyps appear to be well-demarcated lesions variably elevated on a stalk or pedicle. Sessile serrated lesions are flat with indistinct margins, and may show a “mucus cap” that makes them more likely to be missed on optical colonoscopy compared to adenomatous polyps. 

## 4. Screening Tests for LC and CRC, Their Organization and Adhesion

The National Lung Screening Trial in the US demonstrated that, unlike chest X-rays, screening with chest LDCT reduces mortality from LC by 20% in smokers and former smokers [21]. In a recent metanalysis of nine trials, LDCT screening was associated with a 16% relative reduction in LC mortality when compared against a non-screening LDCT control arm [29].

Accordingly, annual LDCT screening of LC is recommended by USPSTF for subjects aged 50–80 years with a smoking history of at least 20 pack years or who have quit in the last 15 years [1]. Experience with LC screening in never smokers is limited to a single study in Asia [30], but the positive results in terms of early stage cancers detection represent an area of further research and debate [31]. Screening in asbestos-exposed workers is effective in detecting asymptomatic LC [32]. Adoption of a validated risk stratification approach is recommended by the European Union (EU) Position Statement to implement LC screening in Europe [33]. 

Several screening tools are available for CRC and its precursor that is the advanced adenoma [34,35,36,37,38]. They include stool-based methods (high-sensitivity guaiac fecal occult blood testing (HSgFOBT) annually, fecal immunochemical testing (FIT) annually and multi-target stool DNA every 1 to 3 years), CT colonography, flexible sigmoidoscopy (FS) and optical colonoscopy (OC), but also barium enema, blood-based tests and colon capsule endoscopy [17]. However, only stool-based methods, CT colonography, flexible sigmoidoscopy and optical colonoscopy have been recommended as screening tools [2,39,40].

So far, the effect of screening in decreasing the CRC incidence and mortality has been demonstrated for stool-based methods [17,41] and for FS [42,43], whereas it is lacking for CT colonography and OC.

According to modeling studies, implementation of screening would yield about a 50% decline in CRC incidence and mortality [44,45]. 

As a matter of fact, the USPSTF recommends screening without identifying a preferred option [2]. In average-risk individuals, recommended screening intervals for CRC depend upon the screening tool and range from one year for stool-based methods to 5 years for CT colonography and FS, to 10 years for OC [2]. In average-risk individuals, the USPSTF recommends CRC screening from 45 to 75 years of age, whereas it can selectively be requested by physicians in subjects aged 76–85 years who had never experienced screening or whose life expectancy is 10 years or more [2]. Advocates of CT colonography [46,47] purport that it is an efficient, underused screening tool for CRC that is intermediate for invasiveness between stool-based methods and OC, while it exhibits a detection rate for advanced adenoma that is higher than stool-based methods and similar to FS or OC. In fact, randomized screening trials showed that CT colonography has higher detection rate for advanced neoplasia (5.2%) than one FIT round (1.7%) [35], and a similar detection rate of flexible sigmoidoscopy [5.1% for CT colonography vs. 4.8% for flexible sigmoidoscopy] [36].

In individuals at high risk of CRC development, it is recommended that screening begins earlier, at age 40 or 10 years before the youngest age of CRC diagnosis in the family [48], and no indication is established concerning the screening tool and interval. However, since OC affords both detection and removal of polyps and adenomas in a single examination, CRC screening in people at high risk must be performed with OC [17,49,50,51] and should be performed every two years or annually [52,53]. 

Screening can follow two basic modalities: opportunistic or organized-population based. Opportunistic screening usually is based on the individual desire to perform search of pre- or early cancer conditions and on an ad hoc or fee-based service, while population-based organized screening is generally supported by the public health service and entails invitation of a target population and measurement and reporting of screening quality [17].

LC screening has been recommended by the USPSTF since 2013 [54], while its implementation as a population-based screening is currently investigated in many European countries. CRC screening using stool-based methods started in around 1996 [55,56] and using OC in 1997 [17], and is predominantly opportunistic in the US, whereas many European countries have implemented population-based CRC screening. 

A general problem of screening initiatives is low adherence, which can jeopardize their efficacy. Adhesion to LC screening is variable and, despite the USPSTF recommendations for annual LDCT screening since 2013, it involved only 17% of the target population in a recent survey in the US, with a non-significant lower participation of non-Hispanic Black individuals [57]. In case of CRC screening, the adherence of average-risk people varies with the screening tool and is higher (55–68%) for FIT and other stool-based modalities [35,37,38], and lower for FS (27–52%) [36,37], CT colonography (25–34%) [34,35,36] and OC (22–35%) [34,38]. Independently from the screening tool, overall adherence to CRC screening in the US is still below 70% in most geographical regions (Centers for Disease Control and Prevention. United States cancer statistics colorectal cancer stat bite. 2020. https://www.cdc.gov/cancer/uscs/about/stat-bites/index.htm, accessed on 30 June 2022). The following interventions to increase the suboptimal CRC screening uptake have been identified: outreach, navigation, education of patients or providers, reminders and financial incentives [17]. It is also conceivable that the opportunity to perform double screening in a single session CT might exert a drag effect on CT colonography in smokers and former smokers undergoing chest LDCT for LC screening, at least in an opportunistic framework.

For both LC and CRC screening, adhesions were reduced for a variety of reasons during the COVID-19 pandemic [58]. In Italy, participation rate in the population-FIT-based CRC screening was 42% in 2019 and 34% in 2020, with a 2019–2020 gap of 1.1 million tests due to the COVID-19 pandemic [59].

## 5. Screening Chest Low-Dose CT and CT Colonography

### 5.1. Operational Aspects

#### 5.1.1. Technical Features

Chest LDCT for LC screening is a simple and fast examination that is obtained with low (typically <40 mAs) current of the radiation tube and thin slice thickness (≤1 mm) during a single breath-hold and without intravenous contrast media administration [60]. The European Society of Thoracic Imaging (ESTI) has set technical standards for LDCT for LC screening (https://www.myesti.org/content-esti/uploads/ESTI-LCS-technical-standards_2019-06-14, accessed on 30 June 2022).

CT colonography for CRC screening is a more complicated procedure than chest LDCT. It requires bowel cathartic preparation using orally administered water solution of macrogol and a low-fiber diet in the few days before examination, and administration of iodinated oral contrast agent for stool tagging 2–3 h before scanning. A reduced cathartic preparation is better tolerated by the subjects and does not affect the results of CT colonography [35,61]. CT colonography is performed after colon distension obtained with an automatic carbon-dioxide insufflator and after intravenous administration of 20 mg of scopolamine [62]. The CT colonography comprises a scout view in a supine position, followed by scanning with slice thickness of 1 mm, generally using a radiation tube current of 50 mAs, from the lung bases to the pelvis. After subject repositioning in the prone position, a second scout view and scan with the same technical parameters from the lung bases to the pelvis are obtained.

#### 5.1.2. Reading the Screening CT Examinations

An experienced radiologist requires less than 10 min to read a LC screening LDCT examination [63]. So far, a double reading of chest LDCT for LC screening, as recommended for breast screening with mammograms, has been advised [60]. To decrease the costs of this procedure, reduce variability in detection rate between readers and overcome shortage of radiologists, experience is rapidly being acquired with test reading by a single radiologist assisted by computer-aided diagnosis (CAD) systems [23,60,64]. ESTI has issued specific recommendations and prepared webinars and workshops for use in CAD support in reading a LC screening test (https://www.myesti.org/lungcancerscreeningcertificationproject/). Moreover, recent studies based on deep learning algorithms for automatic CAD detection of lung nodules yielded very promising results [65,66,67]. Since 2019, ESTI has provided certification courses for education of radiologists in LC screening. 

The reported mean time required to interpret a screening CT colonography performed by an experienced radiologist is 30 min [68]. The time needed to report CT colonography can be substantially reduced using CAD with a “first read double-reading” paradigm, in which the radiologist first examines suspected colonic lesions prompted by CAD and then performs an unassisted reading of the case. This approach reduced the reading time for CT colonography by an experienced radiologist to 4 min without decreasing sensitivity and specificity for polyps equal to or greater than 6 mm [69]. The European Society of Abdominal and Gastrointestinal Radiology (ESGAR) recommends that CT colonography should be read by a specifically trained radiologist with experience in CT image interpretation [70]. Some data indicate that gastro-intestinal radiologists perform better in reading screening CT colonography and that experience in reading at least 1000 CT colonography is associated with a higher detection rate of advanced adenoma [70,71]. However, a recent prospective multicenter study in UK indicated that a 1-day training intervention with performance feedback yielded a cumulative 17% improvement in sensitivity for clinically relevant colorectal neoplasia, including polyps 6 mm or larger and flat lesions, that was independent of previous experience of 80 radiologists [72]. This result opens a path to generalizability and diffusion of the CT colonography as a CRC screening tool [46]. Computer-based self-training systems for CT colonography are also available and can be utilized to improve radiologists’ sensitivity to colonic lesions [73]. 

National quality assurance boards have been advocated to oversee technical standard in LC screening test execution and reading [33].

Quality assurance measures have been proposed for CT colonography (mirroring those of OC) and include techniques of CT colonography, interpretation and outcome measures as colonic and extra-colonic detection rates, rates of missed (interval) cancer and complications [46].

#### 5.1.3. Typical Screening Findings

##### Chest LDCT

LC and precancerous lesions typically appear as a non-calcified pulmonary nodule on LDCT which has a solid, part-solid or non-solid density (Figure 1). Non-nodular presentations of LC are possible and include those associated with cystic airspace, scar-like lesions and perifissural abnormalities [74,75] (Figure 2).

The Lung-RADS 1.1 classification system of the non-calcified lung nodules detected in LDCT has been developed by the American College of Radiology [https://www.acr.org/-/media/ACR/Files/RADS/Lung-RADS/LungRADSAssessmentCategoriesv1-1.pdf?la=en] and is recommended for LC screening practice. It is based on the size of the greatest nodule considering both mean diameter or volume, and it provides instructions for lung nodule management. Notably, the management of screen-detected nodules should be different from that of clinically detected nodules [33] and in the case of Lung-RADS1.1, includes a 6-month follow-up LDCT to measure possible nodule growth for probably benign (category 3) nodules, a 3-month follow up LDCT or Fluoro-Deoxy Glucose Positron Emission Tomography (FDG-PET) CT for suspicious (category 4A) nodules, and FDG-PET and invasive procedures as CT-guided fine needle aspiration or core biopsy and Video-Assisted Thoracic Surgery (VATS) or Robot-Assisted Thoracic Surgery (RATS) for very suspicious (category 4B and 4X) nodules. Management of screen-detected nodules, according to the Lung-RADS 1.1., can be supported by calculators to assess the risk of nodule malignancy as the PAN-CAN (Brock) model [76] (available at https://brocku.ca/lung-cancer-screening-and-risk-prediction/risk-calculators) which take into account additional non-radiological or radiological features, e.g., nodule distribution, shape, etc.. A study comparing five models found that the PAN-CAN model was the most accurate for predicting malignancy of screening detected lung nodules [77].

Additionally, increasing attention is given to computation of volume doubling time as a marker of malignancy [33,60]. According to the European Union Position Statement, management of screen-detected solid lung nodules should use semi-automatically measured volume, when possible, and volume-doubling time [33]. Moreover, baseline nodules greater than 300 mm³ and incident nodules greater than 200 mm³ should be managed in multidisciplinary teams [33].

ESTI has produced a structured report for LC screening containing links for computation of the risk of malignancy of a lung nodule detected at baseline (Brock method) and of the growth at subsequent low-dose CT examinations (www.esti.org).

Recent studies have demonstrated that also deep learning algorithms well predict malignancy of screening detected lung nodules [67,78,79].

##### CT Colonography

In CT colonography, CRC appears as stenosing or vegetating masses (Figure 3), whereas adenomatous polyps appear as small lesions of colonic walls protruding into the colonic lumen, that can be pedunculated or sessile (Figure 3). Flat lesions appear as subtle thickening of colonic walls. 

Artifacts and pseudolesions are common in CT colonography and need to be known by the radiologist reading the screening test [80].

CT Colonography Reporting and Data System (C-RADS) is a method devised to standardize CT colonography reporting [81]. It primarily classifies colonic abnormalities (C). C0 represents an inadequate study. C1 indicates a normal colon or a benign lesion (e.g., Lipoma); in this case, routine screening can be continued. C2 depicts a case in which <3 polyps of 6 to 9 mm in diameter are found, and surveillance with CT colonography or work-up colonoscopy is indicated. C3 represents a case in which a polyp > 9 mm or >3 polyps 6–9 mm in diameter is detected, and the subject has to be referred to work-up colonoscopy. C4 indicates a colonic mass that is likely malignant, and in this case, the surgical referral of the subject is mandatory.

#### 5.1.4. Collateral and Incidental Findings

##### Chest LDCT

Chest LDCT enables demonstration of relevant smoking-related comorbidities in current or former smokers undergoing LC screening. They include pulmonary emphysema [82], interstitial lung disease (ILD) [83] and vascular wall calcifications. These should be labeled as collateral smoking-related findings and should be distinguished from true incidental findings, namely those unrelated to smoking habitude (see below).

In particular, the coronary artery calcifications (CAC) assessed semi-quantitatively using visual scores or quantitatively with the Agatston score are correlated with cardio-vascular (CV) risk factors and increased risk of CV events and death in subjects undergoing LC screening [84,85,86]. Notably, follow-up of subjects recruited in the trials that demonstrated the efficacy of screening with low-dose CT in decreasing mortality from LC have revealed that CV disease is the first cause of non-cancer death in this population [21,22]. This justifies inclusion of the CAC presence and degree in the report of the LDCT examination for LC screening because the subject’s awareness of them could drive changes in lifestyle (primary prevention) [87], prescription and adhesion to statin intake to decrease the CV risk [88,89,90] and ultimately lower the CV events and mortality risks in screened subjects compared with non-screened subjects [91]. 

Detection of ILD in asymptomatic current or former smokers undergoing LC screening is uncommon, but it can have relevant implications for the subject’s health and prognosis following her or his referral to a multidisciplinary specialized team.

Finally, detection of smoking-related comorbidities can yield a more personalized risk stratification with possible influence on the subject’s eligibility and screening regimen (see below) [92].

Incidental findings in chest LDCT are numerous and must be distinguished in those that are actionable, namely that require further investigation and intervention, and those that are non-actionable [60]. The former are less frequent and include neoplastic lesions of the thyroid, thymus, breast, lymph nodes, kidney and liver, as well as aortic artery aneurysm. Non-actionable incidental findings include benign breast lesions such as lipoma or densely calcified nodules, hepatic cysts, renal cysts, etc.

##### CT Colonography

The incidental findings of screening CT colonography are frequent and encompass a large variety of abnormalities, which are unimportant in about 88%, likely unimportant in about 9% and potentially important in 2.5% of cases, respectively [93]. Aside from cancers, the latter include indeterminate renal masses, lymphadenopathy, abdominal aortic aneurysm, obstructing urolithiasis, cirrhosis and sarcoidosis. Osteoporosis, muscle density, fatty liver visceral/subcutaneous fat and abdominal aorta calcifications represent additional biomarkers which can be automatically extracted from screening CT colonography and whose value has been emphasized in view of a personalized opportunistic prediction of future CV events and mortality [94]. 

### 5.2. Radiation Exposures

Some differences in radiation exposure can be observed within current chest LDCT scanning protocols for LC screening, with an upper threshold as high as 2.36 millisievert (mSv) [95] and a mean of 1.2–1.4 mSv in a multicentric study in Italy [96]. For screening CT colonography, the radiation dose comprising the supine and prone acquisitions is usually below 5 mSv [35,97].

Iterative reconstruction of CT images is an alternative to a filtered back-projection method which affords a decrease in the radiation dose. A radiation dose below 1 mSv qualifies for so-called Ultra-Low-Dose Computed Tomography (ULDCT) examinations, which are possible in modern single- or dual-source CT scanners using vendor-specific iterative reconstruction algorithms. So far, ULDCT with iterative reconstruction combined with low radiation tube voltage has been successfully implemented for CT colonography in the clinical and screening setting [98,99,100]. Lung nodule detection in ULDCT examinations has been predominantly investigated in phantom studies [101,102], whereas the experience with clinical or screening lung nodule cases is sparse and initial [103,104,105,106,107], but promising. An alternative approach to improving image quality in ULDCT acquisition is based on deep learning, and it resulted as superior compared to iterative reconstruction in terms of decreasing image noise, increasing nodule detection and ultimately improving measurement accuracy in a recent study in patients outside a screening context [108]. 

ULDCT is ultimately expected to progressively substitute or supplement LDCT for LC screening [33].

### 5.3. Costs

The unitary cost of the chest LDCT examination for LC screening is around EUR/USD 100, but it is likely to decrease once the CAD systems are validated to support the single-radiologist reader approach. 

The unitary cost of CT colonography in the Netherlands and Italy is around EUR 150 [109,110], whereas the cost of OC ranges between EUR 190 (negative examination) and EUR 330 (positive examination including polypectomy and pathology assessment [109,110]. The cost of a single FIT in Italy is about EUR 30 [110].

Analysis of the costs of CT colonography in a trial in Italy [110] has revealed that most of them are related to execution and reading of CT colonography. Cost-containment strategies might include mail invitation for CT colonography to the subject with instructions for collection in a pharmacy of bowel preparation by herself/himself, adoption of a reduced cathartic preparation, containment of operating costs of CT scanners and attainment of full workload for insufflators and reading workstation.

### 5.4. Harms

As for any screening intervention, overdiagnosis is the main harm caused by LC and CRC screening, which can lead to unnecessary procedures and hospitalization.

According to the EU Position Statement [33], a maximum prevalence of 10% of benign resection in lung cancer screening should be attained, while benign resections in clinical trials ranged between 10 to 25% [111,112]. 

The risk of invasive procedure for diagnostic work-up of suspicious pulmonary nodules detected with LDCT screening is low, but not negligible. In fact, major complication rates, including pneumothorax requiring intervention, pulmonary hemorrhage and hemoptysis, are 4.4% for CT-guided Fine Needle Aspiration (FNA) biopsy and 5.7% for CT-guided core biopsy [113]. In turn, the risks of complications of VATS and RATS are 34% and 28%, respectively [114]. 

The risk of bowel perforation or bleeding during CT colonography is negligible, whereas OC, which is the ultimate diagnostic work-up following positive results of stool-based methods or CT colonography, is associated with a very low (below 1%), but not negligible risk of these complications [37,38,115]. Moreover, OC is frequently performed under sedation, and this adds to the overall risk of the procedure.

Another major concern associated with use of CT as a screening examination is the risk of radiation-induced cancer [116,117]. This might be particularly relevant when, as in the case of LDCT screening of LC, the test is repeated annually for a number of years— theoretically from 50 to 80 years of age according to the recent USPSTF guidelines [1], and when one considers that indeterminate screening results require additional examinations as follow-up LDCT, FDG-PET or CT-guided FNA or core biopsy, overall increasing the radiation dose exposure [96]. In particular, studies based on actual dose measurements and the lifetime attributable risk of radiation-induced cancer [118,119,120] have anticipated a small or very small (range 5–0.05%) increase in the risk of radiation-induced solid, especially lung cancers and leukemia in participants of LC screening programs with annual LDCT [121,122,123]. However, several solutions to reduce radiations doses in LDCT screening of LC have been implemented or proposed. These include use of volumetric software to reduce LDCT recall [124]; a biennial rather than annual LDCT schedule, especially in subjects with negative baseline LDCT [125,126]; and ULDCT acquisitions [33,103,104,105,106,107]. 

Similar calculations concerning the risk of radiation-induced cancers can be made for CT colonography if this has to be repeated every 5 years as per the USPSTF recommendation.

### 5.5. Cost Effectiveness

When compared to no screening, both chest LDCT for LC screening [127] and CT colonography for CRC screening [128] are cost effective.

Obviously, biennial screening is more cost effective than annual screening for LC [127], and combination with smoking cessation, which should be offered with screening to current smokers [33], increases the cost effectiveness of LC screening [129]. Areas of active research to further increase the cost effectiveness of LC screening include extension of the 2021 USPSTF recommendation to subjects with different risk profile, i.e., subjects with a 20 pack years history who quit smoking within the past 25 years [130], and personalized screening regimens, with, for instance, biennial rather annual screening or anticipation of screening quit, which takes into account smoking-related comorbidities that are strong competitors of LC as a cause of death [92,131]. Finally, a modeling study indicates that LDCT screening for never smokers might be cost effective in Japan, especially in women, where the LC in never smokers is relatively more frequent, but not in the United States [132]. 

The cost effectiveness of CT colonography has to be compared with that of other screening tools. Accordingly, the higher participation seems to favor CT colonoscopy over OC [109] in average-risk subjects, but the lower participation of CT colonography compared to biennial FIT is a distinctive disadvantage [47]. However, CT colonography is associated with a lower number of work-up OC compared to FIT [47]. An open question with great impact on the cost effectiveness of CT colonography as a population-based CRC screening tool is related to management of the extra-colonic findings [109]. 

## 6. Single Appointment CT for Double LC and CRC Screening

Table 1 summarizes the possible combinations of LC and CRC screening regimens according to the 2021 USPSTF recommendations.

According to the USPSTF, screening of LC with annual chest LDCT is recommended in subjects aged 50 to 80 years who smoked at least 20 cigarettes daily for 20 years (or 10 cigarettes for 40 years, etc.) and quit only in the last 15 years [1]. Screening of CRC with CT colonography every 5 years (or annual stool-based methods, FS every 5 years or OC every 10 years) is recommended in 45–75-year-old subjects with an average risk for CRC, namely those without conditions qualifying as high risk, including familial incidence of CRC, rare inherited diseases as Lynch Syndrome, Familial Adenomatous Polyposis, etc., and inflammatory bowel diseases [2]. Hence, a single CT appointment comprising chest LDCT examination and CT colonography examination affording double LC and CRC screening appears reasonable for subjects of 50–75 years of age with the above smoking history and average CRC risk and who are already suited for opportunistic screening. 

Concerning the possible obstacles to implementation of the single CT appointment for double LC and CRC screening, we identified the following five obstacles. On the radiologist’s side: (1) the considerable amount of time requested (and hence costs) of the double reading, especially without the support of a CAD; (2) the paucity of radiologists skilled in chest LDCT screening for LC and CT colonography for CRC screening; and finally, (3) the tendency of radiologists to be educated and practice in subspecialties and domains in which LC screening with chest LDCT is typically performed by chest radiologists, and CRC screening with CT colonography by gastro-intestinal radiologists. On the screened subject side: (4) the discomfort associated with bowel preparation for CT colonography and (5) “fear of radiations” are the probably the more relevant factors hindering the single appointment for double screening approach. Notably, all these potential obstacles can be addressed: (1) CAD implementation is capable of drastically reducing the reading time of a chest LDCT or CT colonography and improving inter-reader agreement; (2) and (3) promotion of joint initiatives of radiological societies and subsocieties for education in chest LDCT and CT colonography execution and reading. Adoption of a reduced bowel preparation that does not affect detection rate compared to full bowel preparation [35] and improved subject’s communication is likely to overcome obstacles (4) and (5). In particular, in a randomized trial, 88% of subjects who underwent screening CT colonography with reduced cathartic preparation reported no preparation-related symptoms, compared to 70% of those who underwent CT colonography with full bowel preparation [61]. Moreover, no interference of bowel preparation with daily activities was reported in 80% of subjects in the reduced cathartic preparation group compared to 53% of those in the full preparation group [61]. 

Hypothetical implementation of the single CT appointment double screening approach can be envisaged in the opportunistic and population-based frameworks. In the opportunistic framework, CT colonography can reasonably be proposed to smokers or former smokers with average risk for CRC seeking LDCT for LC screening. Within the frame of population-based CRC screening with stool-based methods (or FS), prospective RCT studies can be designed in which invitees with average CRC risk and relevant smoking history are randomized to stool-based methods (or FS) and chest LDCT or to a single appointment CT for double screening with chest LDCT and CT colonography. Additionally, if chest LDCT becomes a population-based preventive intervention for subjects with significant smoking history, as it is expected in the near future in Europe [33], additional CT colonography in a single appointment could be proposed in smokers and former smokers with average CRC.

Notably, all the above possibilities appear justified by the observation that in a recent trial [47], the detection rate of advanced adenoma of CT colonography (5.2%) was higher than that of three biennial FIT (3.1%) in a per-participant analysis, but lower when the analysis was performed considering invitees (1.4% for CT colonography vs. 2.0% for FIT), reflecting the lower participation in CT colonography (26.7%) compared to three biennial FIT (64.9%) [47]. Additionally, subjects with positive CT colonography underwent work-up colonoscopy more frequently than subjects with positive FIT [47].

### 6.1. Operational Aspects

#### 6.1.1. Technical Features

A single appointment CT for chest LDCT and CT colonography must not be confounded with a single CT examination. In fact, colon insufflation can hinder inspiratory expansion of the lung bases with the risk of missing or misinterpretation of findings. Moreover, lung bases in chest LDCT obtained without colon insufflation, namely those obtained annually for LC screening in the target subjects, are difficult to compare with lung bases in chest LDCT obtained after colon insufflation. For these two reasons, we recommend against directly examining the chest and abdomen after colon insufflation. Practically, a single CT session for double LC and CRC screening should entail two examinations (one chest LDCT and one CT colonography) with three separate scouts and corresponding spiral scanning. One scout is obtained in the supine position for the chest, followed by LDCT for LC screening with 40 mAs or less and 1 mm thick slices. After colon insufflation, CT colonography is performed following a second scout in the supine position by abdominal scanning with 50 mAs and 1 mm thick slices. After subject-prone repositioning, the third scout is finally acquired and the abdomen is examined again with the 50 mAs and 1 mm-thick slices spiral scanning.

#### 6.1.2. Reading the Screening CT Examinations

Reading the two screening examinations obtained in a single CT appointment opens scenarios that need to be considered. In fact, in the worst situation concerning availability of reading support technology, i.e., lack of any CAD system, four skilled radiologists and a considerable amount of time are required to provide the double reading (one for chest LDCT and one for CT colonography) traditionally suggested for screening examinations such as a mammography. Availability of CAD systems specific for LC and CRC screening can obviate the double reading and decreases the time required by the radiologists for the single reading. 

An alternative solution to the reading overload is transfer of images to a centralized reading unit equipped with the more advanced reading support technology, where skilled radiologists can perform the test readings. Such a solution is implemented in the Dutch breast cancer screening service (http://www.lrcb.nl/en/breast-cancer, accessed on 30 June 2022) and has already been pursued in some trials of LC and CRC screening trials [23,47].

#### 6.1.3. Typical Screening Findings

The main findings of the single appointment CT for double screening are essentially the same pertaining to chest LDCT and CT colonography alone. 

#### 6.1.4. Collateral and Incidental Findings

The collateral and incidental findings of the single appointment CT for double screening are essentially the same pertaining to chest LDCT and CT colonography alone. 

### 6.2. Radiation Exposure

The cumulative dose delivered in a single CT appointment for double LC and CRC screening comprising three scouts and three spiral acquisitions is expected to be below 7 mSv (about 1.3 mSv for the chest [96] and below 5 mSv for the abdomen [35,97]). 

### 6.3. Costs

The cost of single-appointment CT for a double LC and CRC screening is likely to be EUR/USD 250 or below.

### 6.4. Harms

The potential harms are likely to be same as the chest LDCT and CT colonography alone (see above). 

They include overdiagnosis and risk of intervention for benign lung disease, the risks of invasive procedures for pulmonary nodules work-up, and the risk of bowel perforation or bleeding, especially after the work-up OC after positive CT colonography.

Due to the additive radiation exposure of the chest LDCT and CT colonography, consideration of the risk of radiation-induced cancer is worthy.

### 6.5. Cost-Effectiveness

It is anticipated that in proposing a single CT session for double (LC and CRC) screening to eligible subjects, chest LDCT might exert a positive “drag effect” on adhesion to CT colonography, possibly overcoming the main limitation of the latter; that is, the relatively low participation compared to other CRC screening tools [47]. This possibility indicates that single-appointment CT for double screening approach will deserve a specific cost-effectiveness analysis in due time. 

## 7. Open Issues in CT Screening of LC and CRC

The frequency of screening CT, the impact of iterative reconstruction and deep learning on CT reading and the management of incidental findings are unanswered questions for chest low-dose CT and CT colonography alone. 

The frequency of chest LDCT is annual, but there is evidence that biennial LDCT might not compromise detection of early stage LC, especially in subjects with negative baseline LDCT examination [125,126]. However, so far, no consensus has been reached [33], and USPSTF still recommends annual chest LDCT for LC screening [1]. The recommended frequency of CT colonography in an average-risk individual is every 5 years [2], but no evidence supporting this indication is still available. The above uncertainties have relevant impact on the possible schedules of a single-CT appointment for double LC and CRC screening. In fact, it could range from one double screening CT appointment every five years interleaved with four annual chest LDCT, to one less stringent schedule with one double screening appointment every 6 years interleaved with two biennial chest LDCT. These differences have relevant implications in terms of radiation exposure and costs.

The radiation exposure associated with CT examinations for LC and CRC screening may become harmful in case of repeated CT examinations, especially in the case of annual LDCT for LC screening performed over a 30-year period, as recommended by the USPSTF. Iterative reconstruction of CT images is available in all last-generation CT scanners, and enables substantial radiation dose savings with exposure similar to those of chest X-rays, so called ULDCT. An alternative approach to improving image quality in ULDCT acquisition is based on deep learning [108]. However, the impact of iterative reconstruction or deep learning on reading the LC screening tests has been little explored until now. If chest ULDCT might substitute LDCT for LC screening, this would markedly decrease the overall radiation exposures and risk of radiation-induced cancers [33]. 

Management of incidental findings is a major problem of chest LDCT for LC screening [60], and especially CT colonography for CRC screening [93,109]. Although recommendations about reporting and management of incidental findings are available for LC and CRC screening [60,109], further work is necessary to make them more homogeneous and cost sustainable. 

Obviously, overcoming the above uncertainties will have a major impact on the cost effectiveness of both chest low-dose CT and CT colonography alone, and especially that of the single appointment CT intervention for double screening.

## 8. Conclusions

A single CT appointment for double screening of LC and CRC with chest LDCT followed by CT colonography in a single session is feasible and reasonable as an opportunistic intervention in 50–75-year-old subjects with a definite smoking history and average CRC risk.

Although a definite role for CAD implementation is expected to decrease the screening tests’ reading time and obviate the need for double reading, education of radiologists skilled in both chest LDCT for LC and CT colonography for CRC screening appears fundamental for the single CT appointment double screening scenario proposed here.

Solving persistent uncertainties concerning chest LDCT and CT colonography, including those related to schedule of the CT examinations, role of ULDCT, and management of incidental findings, make it necessary to perform new cost-effectiveness analyses for chest LDCT for LC and CT colonography for CRC screening alone and, especially, for the single CT appointment for double LC and CRC screening.

## Figures and Tables

**Figure 1 diagnostics-12-02326-f001:**
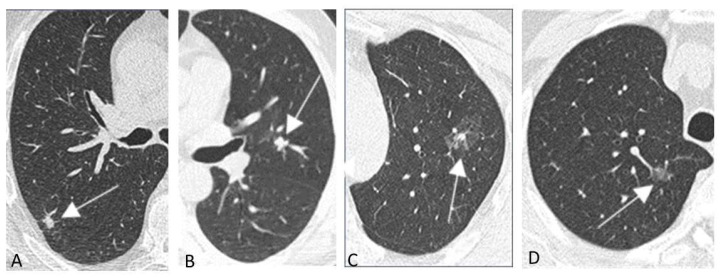
(**A**–**D**) Nodular presentations of early stage lung cancer and precancerous lesions in baseline chest low-dose CT screening in four subjects. Lung cancers (arrows) appearing as solid [(**A**) stage I adenocarcinoma; (**B**) stage I squamous cell carcinoma] or part-solid (**C**) stage I adenocarcinoma) nodules. Atypical adenomatous hyperplasia (arrow) appearing as a pure ground glass nodular opacity (**D**).

**Figure 2 diagnostics-12-02326-f002:**
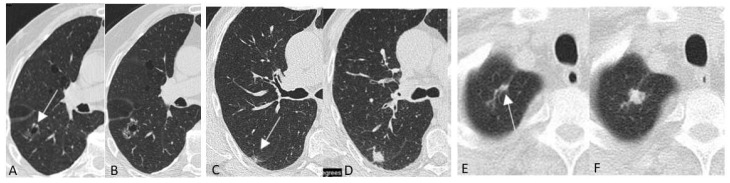
(**A**–**F**) Non-nodular presentations of lung cancer and precancerous lesions in chest low-dose CT screening in three subjects. Lung cancer associated with cystic airspace (arrow) at baseline LDCT (**A**) and two years later (**B**); lung cancer presenting as perifissural abnormality (arrow) at baseline LDCT (**C**) and appearing as solid nodule 2 years later (**D**); lung cancer presenting as scar-like abnormality (arrow) at baseline LDCT (**E**) and appearing as a solid nodule 1 year later (**F**). All images reproduced modified from Mascalchi et al. [74].

**Figure 3 diagnostics-12-02326-f003:**
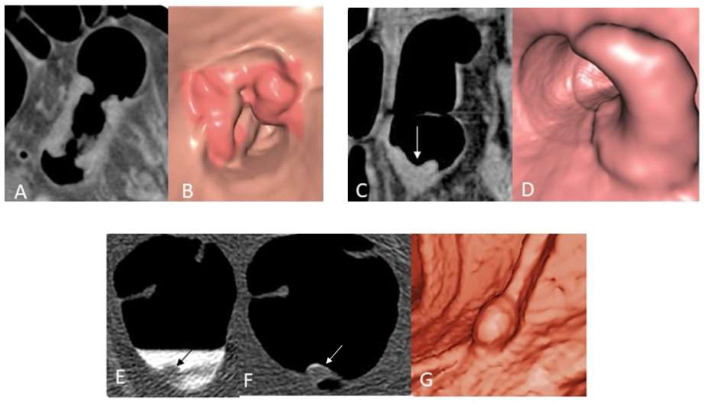
(**A**–**G**) Colon cancer and polyp presentation in CT colonography in three subjects. Stenosing colon cancer in 2D (**A**) and 3D (virtual endoscopy) (**B**) images in subject 1. Vegetating colon cancer (arrow) in 2D (**C**) and 3D (virtual endoscopy) (**D**) images in subject 2. Colon polyp (arrow) in source 2D images obtained in supine (**E**) and prone (**F**) position and in a 3D (virtual endoscopy) (**G**) image in subject 3.

**Table 1 diagnostics-12-02326-t001:** CRC and LC screening regimens in subjects 50–75 years old according to the 2021 USPSTF recommendations.

	Never Smokers	**Smoker/Ex-Smoker ***	Smoker/Ex-Smoker *	Never Smoker
Age (Years)	Average CRC risk °	**Average CRC Risk °**	High CRC Risk	High CRC Risk
50	SBM ^^^ FS ^§^ CTC ^$^ OC ^&^	**chest CT and CTC**	chest CT OC	OC
51	SBM	**chest CT**	chest CT	
52	SBM	**chest CT**	chest CT OC	OC
53	SBM	**chest CT**	chest CT	
54	SBM	**chest CT**	chest CT OC	OC
55	SBM FS CTC	**chest CT and CTC**	chest CT	
56	SBM	**chest CT**	chest CT OC	OC
57	SBM	**chest CT**	chest CT	
58	SBM	**chest CT**	chest CT OC	OC
59	SBM	**chest CT**	chest CT	
60	SBM FS CTC OC	**chest CT and CTC**	chest CT OC	OC
51	SBM	**chest CT**	chest CT	
61	SBM	**chest CT**	chest CT OC	OC
62	SBM	**chest CT**	chest CT	
63	SBM	**chest CT**	chest CT OC	OC
64	SBM	**chest CT**	chest CT	
65	SBM FS CTC	**chest CT and CTC**	chest CT OC	OC
66	SBM	**chest CT**	chest CT	
67	SBM	**chest CT**	chest CT OC	OC
68	SBM	**chest CT**	chest CT	
69	SBM	**chest CT**	chest CT OC	OC
70	SBM FS CTC OC	**chest CT and CTC**	chest CT	
71	SBM	**chest CT**	chest CT OC	OC
72	SBM	**chest CT**	chest CT	
73	SBM	**chest CT**	chest CT OC	OC
74	SBM	**chest C** **T**	chest CT	
75	SBM FS CTC	**chest CT and CTC**	chest CT OC	OC

* LC screening is recommended in subjects of 50–80 years with at least 20 pack years of smoking history and who have quit in the last 15 years. ° Average CRC-risk subjects include those without familial history of the disease, rare inherited diseases as Lynch Syndrome and Familial Adenomatous Polyposis, and inflammatory bowel diseases such as ulcerative colitis and Crohn’s colitis, who qualify as high-risk CRC subjects. ^^^ SBM = stool-based methods, including high-sensitivity guaiac fecal occult blood testing, fecal immunochemical testing and multi-target stool DNA; ^§^ FS = flexible sigmoidoscopy; ^$^ CTC = CT colonography; ^&^ OC = optical colonoscopy.

## Data Availability

The data presented in this review articles can be retrieved from the references detailed below and from the www addresses indicated in the text.

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
