# Peer review of "Single CT Appointment for Double Lung and Colorectal Cancer Screening: Is the Time Ripe?"

_diagnostics, 2022, doi:10.3390/diagnostics12102326_

Round 1

Reviewer 1 Report

Lung cancer(LC) and colorectal cancer(CRC) are two of the most common and lethal tumors worldwide. CT screening of LC and CRC helps in the early detection and treatment of tumors, and reduces mortality. The authors based on a retrospective study to explore the possibility of combining low-dose CT of the chest and CT colonoscopy for dual LC and CRC screening in a single CT appointment. This is an intriguing topic for the researchers involved. However, the paper needs very significant improvement before acceptance for publication.

Major comments:

1. The sections on "Epidemiology" and "Pathology" are a bit confusing, lack focus, and are not closely related to the main topic. The author should streamline the sections and highlight advances in the use of CT for LC and CRC screening.

2. Literature evidence should be reasonably summarized and analyzed when cited, and its potential causes and feasible solutions can be appropriately elaborated, rather than simply describing relevant results and data, lacking reasonable language organization and inconveniencing the reader.

3. In the part of “Single appointment CT for double LC and CRC screening”, the authors should cite reliable evidence to enhance the credibility of the hypothesis and analysis, and the part should be further divided into points for the reader's reading.

4. Some of the viewpoints mentioned in this paper are clichéd and do not link them well to the main theme. More constructive and innovative specific ideas should be proposed for the theme.

Minor comments:

1. Figures are unsatisfactory, some of the pictures are missing arrows, please make changes to make them neat and beautiful.

2. There is a redundant row in the table about “the age of 51”, and the meaning of the special symbols and abbreviations appearing in the table should be stated in the table notes.

3. The current article has many spelling and citation formatting errors, please check carefully and revise.

4. This article needs further language editing to make the content more fluent and easy to understand.

Author Response

Reviewer 1

Major comments:

  1. The sections on "Epidemiology" and "Pathology" are a bit confusing, lack focus, and are not closely related to the main topic. The author should streamline the sections and highlight advances in the use of CT for LC and CRC screening.

Re: The sections were revised. In particular, the one dealing with epidemiology of LC was implemented with quotations of papers demonstrating the improved efficiency of selection criteria based on risk calculators as compared to the simple “age and pack years” criteria. The following text and references were added.

Notably, several studies have demonstrated that selection of subjects to be invited to LC screening based on age and pack-years only, as indicated by the USPSTF, perform worst in predicting LC risk, and hence are suboptimal for subjects selection, as compared to personal risk prediction calculation models which consider additional risk factors as history of respiratory diseases, previous malignancy, family history of lung cancer (first-degree relative diagnosed at age 60 years or younger), and exposure to asbestos [13,14]. So far, the modified Liverpool Lung Project [LLPv2] and Prostate, Lung, Colorectal and Ovarian Cancer Screening Trial [PLCO] models [15,16] have been those more frequently utilized to select patients for LC screening in a clinical trial.

New references

  1. Li K et al. Selecting High-Risk Individuals for Lung Cancer Screening: A Prospective Evaluation of Existing Risk Models and Eligibility Criteria in the German EPIC Cohort. Cancer Prev Res (Phila) 2015, 8, 777-785.
  2. Tammemägi et al. USPSTF2013 versus PLCOm2012 lung cancer screening eligibility criteria (International Lung Screening Trial): interim analysis of a prospective cohort study. Lancet Oncol 2022, 23, 138-148.
  3. Cassidy et al. The LLP risk model: an individual risk prediction model for lung cancer. Br J Cancer 2008, 98, 270-276.
  4. Bach et al. Variations in lung cancer risk among smokers. J Natl Cancer Inst  2003, 95, 470-478.

  1. Literature evidence should be reasonably summarized and analyzed when cited, and its potential causes and feasible solutions can be appropriately elaborated, rather than simply describing relevant results and data, lacking reasonable language organization and inconveniencing the reader.

Re: Where possible this was done by expanding the corresponding text. 

  1. In the part of “Single appointment CT for double LC and CRC screening”, the authors should cite reliable evidence to enhance the credibility of the hypothesis and analysis, and the part should be further divided into points for the reader's reading.

Re: To the best of our knowledge, despite its reasonability, the single CT appointment for double screening has not proposed or pursued yet.

As requested, the section was divided in the same way of the prior sections dealing with LC and CRC screening alone, and some material was displaced in the new sections.

  1. Some of the viewpoints mentioned in this paper are clichéd and do not link them well to the main theme. More constructive and innovative specific ideas should be proposed for the theme.

Re:  We thank the Reviewer for this input and strived to follow it.

  1. A) The aspects concerning reading of the CT screening examinations were revised as follows:
  2. Screening chest low dose CT and CT colonography

4.1 Operational aspects

Reading the screening CT examinations

An experienced radiologist requires less than 10 minutes for reading a LC screening LDCT examination [63]. So far, a double reading of chest LDCT for LC screening, as recommended for breast screening with mammograms, has been advised [60]. To decrease costs of this procedure, reduce variability in detection rate between readers, and overcome shortage of radiologists, experience is rapidly being acquired with test reading by a single radiologist assisted by computer-aided diagnosis (CAD) systems [23,60,64]. ESTI has issued specific recommendations as well as prepared webinars and workshops for use of CAD support in reading LC screening test (https://www.myesti.org/lungcancerscreeningcertif​icationproject/ ). Moreover, recent studies based on deep learning algorithms for automatic CAD detection of lung nodules yielded very promising results [65-67]. Since 2019 ESTI provides certification courses for education of radiologists to LC screening.

The reported mean time required to interpret a screening CT colonography by an experienced radiologist is 30 minutes [68]. The time needed to report CT colonography can be substantially reduced using CAD with a “first read double-reading” paradigm, in which the radiologist first examines suspected colonic lesions prompted by CAD and then performs an unassisted reading of the case. This approach reduced the reading time for CT colonography by an experienced radiologist to 4 minutes without decreasing sensitivity and specificity for polyps equal to or greater than 6 mm [69]. The European Society of Abdominal and Gastrointestinal Radiology recommends that CT colonography should be read by a specifically trained radiologist with experience in CT image intrepretation [70]. Some data indicate that gastro-intestinal radiologists perform better in reading screening CT colonography and that experience in reading at least 1000 CT colonography is associated with a higher detection rate of advanced adenoma [70, 71]. However, a recent prospective multicenter study in UK indicated that a 1-day training intervention with performance feedback yielded a cumulative 17% improvement in sensitivity for clinically relevant colorectal neoplasia, including polyps 6 mm or larger and flat lesions, that was independent of previous experience of 80 reading radiologists [72]. This result opens the way to generalizability and diffusion of the CT colonography as a CRC screening tool [46]. Computer-based self-training systems for CT colonography are also available and can be utilized to improve radiologists' sensitivity for colonic lesions. [73].

National quality assurance boards have been advocated to oversee technical standard in LC screening test execution and reading [33].

Quality assurance measures have been proposed for CT colonography (mirroring those of OC) and include techniques of CT colonography, interpretation, and ouctome measures as colonic and extra-colonic detection rates, rates of missed (interval) cancer, and complications [46].

New references

  1. Oudkerk et al. European position statement on lung cancer screening. Lancet Oncol 2017, 18, e754-e766.
  2. Pickhardt, P.J. CT Colonography: The Role of Radiologist Training. Radiology 2022, 303, 371-372.
  3. Hwang et al. Variability in interpretation of low-dose chest CT using computerized assessment in a nationwide lung cancer screen ing program: comparison of prospective reading at individual institutions and retrospective central reading. Eur Radiol 2021, 31,2845–2855.
  4. Huang et al. Added Value of Computer-aided CT Image Features for Early Lung Cancer Diagnosis with Small Pulmonary Nodules: A Matched Case-Control Study. Radiology 2018, 286, 286-295.
  5. Cui et al. Development and clinical application of deep learning model for lung nodules screening on CT images. Sci Rep 2020, 10, 13657.
  6. Cui et al. Performance of a deep learning-based lung nodule detection system as an alternative reader in a Chinese lung cancer screening program. Eur J Radiol 2022, 146, 110068.
  7. Shao et al. Deep Learning Empowers Lung Cancer Screening Based on Mobile Low-Dose Computed Tomography in Resource-Constrained Sites. Front Biosci (Landmark Ed) 2022, 27, 212.
  8. Obaro et al. Computed tomographic colonography: how many and how fast should radiologists report? Eur Radiol 2019, 29,5784-5790
  9. Neri et al. The second ESGAR consensus statement on CT colonography. Eur Radiol 2013, 23, 720-729.
  10. Plumb et al. Use of CT colonography in the English Bowel Cancer Screening Programme. Gut 2014, 63, 964-973
  11. Obaro et al. Colorectal Cancer: Performance and Evaluation for CT Colonography Screening- A Multicenter Cluster-randomized Controlled Trial. Radiology 2022, 303(361-370. doi: 10.1148/radiol.211456. Epub 2022 Feb 15.
  12. Sali et al. Computer-based self-training for CT colonography with and without CAD. Eur Radiol 2018, 28, 4783-4791.

  1. B) Moreover, we attempted to outline the possible obstacles to the single CT appointment double screening approach.

The following were identified as possible obstacles on the radiologists side: 1) the considerable amount of time requested (and hence costs) of the double reading, especially without the support of a CAD; 2)  the paucity of radiologists skilled in chest LDCT screening for LC and CT colonography for CRC screening, and finally, 3)  the tendency of radiologists to be educated and practice in subspecialties domains in which LC screening with chest LDCT is typically performed by chest radiologists, and CRC screening with CT colonography by gastro-intestinal radiologists. On the screened subject side, 4) the discomfort associated with bowel preparation for CT colonography and 5) “fear of too much radiations” are the probably the more relevant factors hindering the single appointment for double screening approach.

Notably, all these potential obstacles can be addressed: 1) CAD implementation is capable to drastically reduce the reading time of a chest LDCT or CT colonography and to improve inter-reader agreement; 2) and 3) promotion of joint initiatives of radiological societies and subsocietiess for education in chest LDCT and CT colonography execution and reading; improved subject’communication is likely to overcome obstacles 4) and 5).

The text was modified as follows:

  1. Single appointment CT for double LC and CRC screening

Concerning the possible obstacles to implementation of the single CT appointment for double LC and CRC screening we identified the following five. On the radiologists side: 1) the considerable amount of time requested (and hence costs) of the double reading, especially without the support of a CAD; 2)  the paucity of radiologists skilled in chest LDCT screening for LC and CT colonography for CRC screening, and finally, 3)  the tendency of radiologists to be educated and practice in subspecialties domains in which LC screening with chest LDCT is typically performed by chest radiologists, and CRC screening with CT colonography by gastro-intestinal radiologists. On the screened subject side: 4) the discomfort associated with bowel preparation for CT colonography and 5) “fear of radiations” are the probably the more relevant factors hindering the single appointment for double screening approach. Notably, all these potential obstacles can be addressed: 1) CAD implementation is capable to drastically reduce the reading time of a chest LDCT or CT colonography and to improve inter-reader agreement; 2) and 3) promotion of joint initiatives of radiological societies and subsocieties for education in chest LDCT and CT colonography execution and reading. Adoption of a reduced bowel preparation that does not affect detection rate as compared to full bowel preparation [35] and improved subject’s communication are likely to overcome obstacles 4) and 5). In particular, in a randomised trial, 88% of subjects who underwent screening CT colonography with reduced cathartic preparation reported no preparation-related symptoms, as compared to 70% of those who underwent CT colonography with full bowel preparation [61]. Moreover, no interference of bowel preparation with daily activities was reported in 80% of subjects in the reduced cathartic preparation group as compared to 53% of those in the full preparation group. [61].

  1. C) We finally provided some hypotheses about the implementation of the single appointment double screening approach as follows:

  1. Single appointment CT for double LC and CRC screening

Hypothetical implementation of the single CT appointment double screening approach can be envisaged in the opportunistic and population-based frameworks. In the opportunistic framework, CT colonography can reasonably be proposed to smokers or former smokers with average risk for CRC seeking LDCT for LC screening. Within the frame of population-based CRC screening with stool based methods (or FS), prospective RCT studies can be designed in which invitees with average CRC risk and a relevant smoking history are randomized to stool based methods (or FS) and chest LDCT or to a single appointment CT for double screening with chest LDCT and CT colonography. Also, if chest LDCT becomes a population based preventive intervention for subjects with significant smoking history, as it is expected in the near future in Europe [33], additional CT colonography in a single appointment could be proposed in smokers and former smokers with average CRC.

Notably, all the above possibilities appear justified by the observation that in a recent trial [47] the detection rate of advanced adenoma of CT colonography (5.2%) was higher than that of 3 biennial FIT (3.1%) in a per- participant analysis, but lower when the analysis was performed considering invitees (1.4% for CT colonography vs 2.0% for FIT), reflecting the lower participation in CT colonography (26.7%) as compared to 3 biennial FIT (64.9%) [47]. Also, subjects with positive CT colonography underwent work-up colonoscopy more frequently than subjects with positive FIT [47].

Minor comments:

  1. Figures are unsatisfactory, some of the pictures are missing arrows, please make changes to make them neat and beautiful.

Re:  The figures were revised.

  1. There is a redundant row in the table about “the age of 51”, and the meaning of the special symbols and abbreviations appearing in the table should be stated in the table notes.

Re: We apologize for the mistake in Table 1. The redundant row was corrected. The meaning of the special symbols and abbreviations were stated in the bottom table legend.

  1. The current article has many spelling and citation formatting errors, please check carefully and revise.

Re: The manuscript was checked for the many spelling and citation errors. References were modified according to Diagnostics instructions.

  1. This article needs further language editing to make the content more fluent and easy to understand.

Re: Language editing was performed.

Reviewer 2 Report

general comments

This article presents the possibility of performing lung cancer screening and colorectal cancer screening at the same time. This review is informative and valuable. However, some interpretations are not valid, so the article needs minor corrections. I think you need at least the following fixes:

main point

1. Authors should demonstrate diagnostic capability to date for LC and CRC. By indicating the level of diagnostic ability, I think the claim of this paper becomes clear.

2. In recent years, deep learning has begun to be used for diagnostic imaging. Applying deep learning to image reconstruction and diagnostic imaging has the potential to provide more accurate examinations at lower cost. I think it is necessary to explain these possibilities.

minor point

1. 3. LC and CRC screening tests, their composition and adhesion

I believe a barium enema is included in CRC screening.

2. 4. Screening chest low-dose CT and CT colonography: How to read a screening CT study

In Japan and other countries, there is a shortage of radiologists and reading can be unwieldy.

3. Figure 1, Figure 2

The position of the arrow is confusing and needs to be fixed.

It is difficult to distinguish between the same case and different cases, so it is necessary to devise ways to publish it.

4. 5. Single appointment CT for double LC and CRC screening: Line 440-

Is it possible to do LC screening and CT colonography in the same scout, i.e. can't you scan the chest and abdomen after colon insufflation?

Author Response

Reviewer 2

 main point

  1. Authors should demonstrate diagnostic capability to date for LC and CRC. By indicating the level of diagnostic ability, I think the claim of this paper becomes clear.

Re: We thank the reviewer for this request. The following modifications to the text were made:

  1. Screening tests for LC and CRC, their organization and adhesion

The National Lung Screening Trial in US demonstrated that, differently of chest X-rays, screening with chest LDCT reduces mortality from LC by 20% in smokers and former smokers [21]. In a recent metanalysis of 9 trials, LDCT screening was associated with a 16% relative reduction in LC mortality when compared against a non-screening LDCT control arm [29].

  Accordingly, annual LDCT screening of LC is recommended by USPSTF for subjects aged 50-80 years with a smoking history of at least 20 pack-years who have quit in the last 15 years [1].  Experience with LC screening in never smokers is limited to a single study in Asia [30], but the positive results in terms of early-stage cancers detection represent an area of further research and debate [31]. Screening in asbestos-exposed workers is effective in detecting asymptomatic LC [32]. Adoption of a validated risk stratification approach is recommended by the European Union Position Statement to implement LC screening in Europe [33].

Several screening tools are available for CRC and its precursor that is the advanced adenoma [34-38]. They incude stool based methods [(high sensitivity guaiac fecal occult blood testing (HSgFOBT) annually, fecal immunochemical testing (FIT) annually, and Multi-target stool DNA every 1 to 3 years], CT colonography, flexible sigmoidoscopy (FS) and optical colonoscopy (OC), but also barium enema, blood based tests and colon capsule endoscopy [17]. However, only stool based methods, CT colonography, flexible sigmoidoscopy and optical colonoscopy have been recommended as screening tools [2, 39, 40].

 So far, the effect of screening in decreasing the CRC incidence and mortality has been demonstrated for stool based methods [17, 41] and for FS [42, 43], whereas it lacking for CT colonography and OC.

According to modelling studies, implementation of screening would yield about a 50% decline in CRC incidence and mortality [44, 45] .

As a matter of fact, the USPSTF recommends screening without identifying a preferred option [2]. In average-risk individuals recommended screening intervals for CRC depend upon the screening tool and range from one year for stool based methods, to 5 years for CT colonography and FS, to 10 years for OC [2]. In average-risk individuals the USPSTF recommends CRC screening from 45 to 75 years of age, whereas it can selectively be requested by physicians in subjects aged 76-85 years who had never performed screening or whose life-expectancy is 10 years or more [2]. Advocates of CT colonography [46,47] purport that it is an efficient underused screening tool for CRC that is intermediate for invasiveness between stool based methods and OC, while it exhibits a detection rate for advanced adenoma that is higher than stool based methods ad similar to FS or OC. In fact, randomized screening trials showed that CT colonography has higher detection rate for advanced neoplasia (5.2%) than one FIT round (1.7%) [35], and similar detection rate of flexible sigmoidoscopy [5.1% for CT colonography vs. 4.8% for flexible sigmoidoscopy] [36].

 In individuals at high-risk of CRC development, screening is recommended to begin earlier, namely at age 40 or 10 years before the youngest age of CRC diagnosis in the family [48] and no indication is established concerning the screening tool and interval. However, since OC affords both detection and removal of polyps and adenomas in a single examination, CRC screening in people at high risk must be performed with OC  [17,49-51] and should be performed every two years or annually [52, 53].

(…)

A general problem of screening initiatives is the low adherence which can jepordize their efficacy. Adhesion to LC screening is variable and, despite the USPSTF recommendations for annual LDCT screening since 2013, it involved only 17% of the target population in a recent survey in the US, with a non-significant lower participation of non-Hispanic Black individuals [57]. In case of CRC screening, the adherence of average–risk people vary with the screening tools and is higher (55-68%) for FIT and other stool based modalities [35, 37, 38], and lower for FS (27-52%) [36, 37], CT colonography (25-34%) [34-36] and OC (22-35%) [34, 38]. Independently from the screening tool, overall adherence to CRC screening in US is still below 70% in most geographical regions [Centers for Disease Control and Prevention. United States cancer statistics colorectal cancer stat bite. 2020. https://www.cdc.gov/ cancer/uscs/about/stat-bites/index.htm]. The following interventions to increase the suboptimal CRC screening uptake have been identified: outreach, navigation, education of patients or providers, reminders, and financial incentives [17]. It is conceivable that also the opportunity to perform double screening in a single session CT migh exert a drag effect on CT colonography in smokers and former smokers undergoing chest LDCT for LC screening, at least in an opportunistic framework.

New references

  1. Field et al. Lung cancer mortality reduction by LDCT screening: UKLS randomised trial results and international meta-analysis. Lancet Reg Health Eur 2021, 10, 100179.
  2. European Union. Council recommendation of 2 December 2003 on cancer screening. 2003. https://eurlex.europa.eu/LexUriServ/ LexUriServ.do?uri=OJ:L:2003:327:0034:0038:EN:PDF (accessed September 7, 2022)
  3. Wolf et al. Colorectal cancer screening for average-risk adults: 2018 guideline update from the American Cancer Society. CA Cancer J Clin 2018, 68, 250-281.
  4. Ventura et al.. The impact of immunochemical faecal occult blood testing on colorectal cancer incidence. Dig Liver Dis 2014, 46, 82-86.
  5. Atkin et al. Once-only flexible sigmoidoscopy screening in prevention of colorectal cancer: a multicentre randomised controlled trial. Lancet 2010, 375,1624-33.
  6. Meester et al. Colorectal cancer deaths attributable to nonuse of screening in the United States. Ann Epidemiol2015, 25, 208-213.e1.
  7. Zauber, AG. The impact of screening on colorectal cancer mortality and incidence: has it really made a difference? Dig Dis Sci2015, 60, 681-91. 

  1. In recent years, deep learning has begun to be used for diagnostic imaging. Applying deep learning to image reconstruction and diagnostic imaging has the potential to provide more accurate examinations at lower cost. I think it is necessary to explain these possibilities.

Re: We thank the Reviewer for this input and performed an ad-hoc search on PubMed.

Studies based on deep learning for LC screening with CT focused on two aspects: automatic detection of lung nodules (Cui et al. 2020; Cui et al. 2022; Jiang et al. 2022), capability to predict malignancy of detected nodules (Ciompi et al. 2017; Venkadesh et al. 2021) or both (Shao et al. 2022).

This was mentioned in the current version of the paper as follows:

  1. Screening chest low dose CT and CT colonography

4.1 Operational aspects

Reading the CT examination

(…)

Moreover, recent studies based on deep learning algorithms for LC screening with CT focused on automatic detection of lung nodules with very promising results [65-67].

Typical screening findings

                (..)

Recent studies have demonstrated the capability of deep learning algorithms to predict malignancy of detected nodules [67, 78, 79].

4.2 Radiation exposures

An alternative approach to improve image quality in ULDCT acquistion is based on deep learning and it resulted superior as compared to iterative reconstruction in decreasing image noise, increasing nodule detection and ultimately impoving measurement accuracy in a recent study in patients outside a screening context [118]. 

New references

  1. Cui et al. Development and clinical application of deep learning model for lung nodules screening on CT images. Sci Rep 2020 ,10, 13657.
  2. Cui et al. Performance of a deep learning-based lung nodule detection system as an alternative reader in a Chinese lung cancer screening program. Eur J Radiol 2022 ,146,110068.
  3. Shao et al. Deep Learning Empowers Lung Cancer Screening Based on Mobile Low-Dose Computed Tomography in Resource-Constrained Sites. Front Biosci (Landmark Ed) 2022, 27, 212.
  4. Ciompi et al. Towards automatic pulmonary nodule management in lung cancer screening with deep learning. Sci Rep 2017, 7, 46479.
  5. Venkadesh et al. Deep Learning for Malignancy Risk Estimation of Pulmonary Nodules Detected at Low-Dose Screening CT. Radiology 2021,300, 438-447
  6. Jiang et al. Deep Learning Reconstruction Shows Better Lung Nodule Detection for Ultra-Low-Dose Chest CT. Radiology 2022, 303, 202-212.

In the realm of CRC screening with CT colonography, we found that deep learning has recently been applied to the differentiation between benign and premalignant polyps [Wesp et al. Deep learning in CT colonography: differentiating premalignant from benign colorectal polyps. Eur Radiol. 2022 Jul;32(7):4749-4759.  

We did not feel that such an application is worthy of citation in a general review paper as ours and skipped it.

minor point

  1. LC and CRC screening tests, their composition and adhesion

Re: Composition and adhesion to LC and CRC screening tests were revised.

Introduction

However, importantly, while chest LDCT is the only recommended screening tool for LC since 2013, other screening tools besides CT colonography have been as well recommended for CRC since late 90’ [1,2].

  1. Screening tests for LC and CRC, their organization and adhesion

(…)

Several screening tools are available for CRC and its precursor that is the advanced adenoma [34-38]. They incude stool based methods [(high sensitivity guaiac fecal occult blood testing (HSgFOBT) annually, fecal immunochemical testing (FIT) annually, and Multi-target stool DNA every 1 to 3 years], CT colonography, flexible sigmoidoscopy (FS) and optical colonoscopy (OC), but also barium enema, blood based tests and colon capsule endoscopy [17]. However, only stool based methods, CT colonography, flexible sigmoidoscopy and optical colonoscopy have been recommended as screening tools [2, 39, 40].

(…)

A general problem of screening initiatives is the low adherence which can jepordize their efficacy. Adhesion to LC screening is variable and, despite the USPSTF recommendations for annual LDCT screening since 2013, it involved only 17% of the target population in a recent survey in the US, with a non-significant lower participation of non-Hispanic Black individuals [57]. In case of CRC screening, the adherence of average–risk people vary with the screening tools and is higher (55-68%) for FIT and other stool based modalities [35, 37, 38], and lower for FS (27-52%) [36, 37], CT colonography (25-34%) [34-36] and OC (22-35%) [34, 38]. Independently from the screening tool, overall adherence to CRC screening in US is still below 70% in most geographical regions [Centers for Disease Control and Prevention. United States cancer statistics colorectal cancer stat bite. 2020. https://www.cdc.gov/ cancer/uscs/about/stat-bites/index.htm]. The following interventions to increase the suboptimal CRC screening uptake have been identified: outreach, navigation, education of patients or providers, reminders, and financial incentives [17]. It is conceivable that also the opportunity to perform double screening in a single session CT migh exert a drag effect on CT colonography in smokers and former smokers undergoing chest LDCT for LC screening, at least in an opportunistic framework.

For both LC and CRC screening, adhesions were reduced for a variety of reasons during the COVID-19 pandemic [58]. In Italy, participation rate in the population-FIT-based CRC screening was 42% in 2019 and 34% in 2020 with a 2019-2020 gap of 1.1 million tests due to COVID-19 pandemic [59].

  1. 4. Screening chest low-dose CT and CT colonography: How to read a screening CT study

 In Japan and other countries, there is a shortage of radiologists and reading can be unwieldy.

Re: We tried to emphasize the pivotal role of CAD systems as a reliable way to overcome the double reading and the shortage of radiologists. The text was modified as follows:

Reading the screening CT examinations

An experienced radiologist requires less than 10 minutes for reading a LC screening LDCT examination [63]. So far, a double reading of chest LDCT for LC screening, as recommended for breast screening with mammograms, has been advised [60]. To decrease costs of this procedure, reduce variability in detection rate between readers, and overcome shortage of radiologists, experience is rapidly being acquired with test reading by a single radiologist assisted by computer-aided diagnosis (CAD) systems [23,60,64]. ESTI has issued specific recommendations as well as prepared webinars and workshops for use of CAD support in reading LC screening test (https://www.myesti.org/lungcancerscreeningcertif​icationproject/ ). Moreover, recent studies based on deep learning algorithms for automatic CAD detection of lung nodules yielded very promising results [65-67]. Since 2019 ESTI provides certification courses for education of radiologists to LC screening.

The reported mean time required to interpret a screening CT colonography by an experienced radiologist is 30 minutes [68]. The time needed to report CT colonography can be substantially reduced using CAD with a “first read double-reading” paradigm, in which the radiologist first examines suspected colonic lesions prompted by CAD and then performs an unassisted reading of the case. This approach reduced the reading time for CT colonography by an experienced radiologist to 4 minutes without decreasing sensitivity and specificity for polyps equal to or greater than 6 mm [69]. The European Society of Abdominal and Gastrointestinal Radiology recommends that CT colonography should be read by a specifically trained radiologist with experience in CT image intrepretation [70]. Some data indicate that gastro-intestinal radiologists perform better in reading screening CT colonography and that experience in reading at least 1000 CT colonography is associated with a higher detection rate of advanced adenoma [70, 71]. However, a recent prospective multicenter study in UK indicated that a 1-day training intervention with performance feedback yielded a cumulative 17% improvement in sensitivity for clinically relevant colorectal neoplasia, including polyps 6 mm or larger and flat lesions, that was independent of previous experience of 80 reading radiologists [72]. This result opens the way to generalizability and diffusion of the CT colonography as a CRC screening tool [46]. Computer-based self-training systems for CT colonography are also available and can be utilized to improve radiologists' sensitivity for colonic lesions. [73].

National quality assurance boards have been advocated to oversee technical standard in LC screening test execution and reading [33].

Quality assurance measures have been proposed for CT colonography (mirroring those of OC) and include techniques of CT colonography, interpretation, and ouctome measures as colonic and extra-colonic detection rates, rates of missed (interval) cancer, and complications [46].

  1. Figure 1, Figure 2

 The position of the arrow is confusing and needs to be fixed.

Re: Larger arrows were adopted and replaced the former in the images of new Fig.1, Fig.2 and Fig.3 (see below).

It is difficult to distinguish between the same case and different cases, so it is necessary to devise ways to publish it.

Re: To make it easier to recognize which images belong to the same patient we grouped the panels accordingly in the Figures. Moreover, to distinguish between nodular LC detected at baseline LDCT from non-nodular presentation of LC we splitted the original Fig.1 in two new panels named Fig.1 (LC detected and diagnosed at baseline LDCT) and Fig. 2(LC presenting as non-nodular lesions which were diagnosed because of growth at subsequent LDCT). Old Fig. 2 has become new Fig. 3. The images and legends were modified a as follows:

Fig. 1 A-D) Nodular presentations of earlu stage lung cancer and precancerous lesions in baseline chest low dose CT screening in four subjects. Lung cancers (arrows) appearing as solid (A, stage I adenocarcinoma; B, stage I squamous cell carcinoma) or part-solid (C, stage I adenocarcinoma) nodules. Atypical adenomatous hyperplasia (arrow) appearing as a pure ground glass nodular opacity (D).

Fig. 2 A-F) non-nodular presentations of lung cancer and precancerous lesions in chest low dose CT screening in three subjects. Lung cancer associated with cystic airspace (arrow) at baseline LDCT (A) and two years later (B); lung cancer presenting as perifissural abnormality (arrow) at baseline LDCT (C) and appearing as solid nodule 2 years later (D); lung cancer presenting as scar-like abnormality (arrow) at baseline LDCT (E) and appearing as a solid nodule 1 year later (F). All images reproduced modified from Mascalchi et al. [74].

Fig. 3 A-G) Colon cancer and polyp presentation in CT colonography in three subjects. Stenosing colon cancer in 2D (A) and 3D (virtual endoscopy) (B) images in subject 1. Vegetating colon cancer (arrow) in 2D (C) and 3D (virtual endoscopy) (D) image in subject 2. Colon polyp (arrow) in source 2D images obtained in supine (E) and prone (F) position and in a 3D (virtual endoscopy) (G) image in subject 3.

  1. 5. Single appointment CT for double LC and CRC screening: Line 440-

Is it possible to do LC screening and CT colonography in the same scout, i.e. can't you scan the chest and abdomen after colon insufflation?

RE: We advised against such a procedure. It should be avoided because: 1) obliteration of pleural recesses hinders evaluation of corresponding lung areas, and 2) lack of matching with intervening annual LDCT examinations for LC screening obtained withoust colonic insufflation.

The text was modified as follows:

5.1 Operational aspects

Technical features

Single appointment CT for chest low dose CT and CT colonography must not be confounded with a single CT examination. In fact, since colon insufflation can hinder inspiratory expansion of the lung bases with the risk of missing or misinterpretation of findings. Moreover, lung bases in chest low dose CT obtained without colon insufflation, namely those obtained annually for LC screening in the target subjects, are difficult to compare with lung bases in chest low dose CT obtained after colon insufflation. For these two reasons, we recommend against examining directly the chest and abdomen after colon insufflation. Practically, the single CT session for double LC and CRC screening should entail two examinations (one chest Low dose CT and one CT colonography) with overall three separate scouts and corresponding spiral scanning. One scout is obtained in the supine position for the chest, followed by low dose CT for LC screening with 40mAs or less and 1 mm thick slices.  After colon insufflation, CT colonography is performed following a second scout in the supine position by abdominal scanning with 50mAs and 1 mm thick slices. After subject prone repositioning, the third scout is finally acquired and the abdomen again examined with the with 50mAs and 1 mm thick slices spiral scanning.

Round 2

Reviewer 1 Report

none